# Dietary Habits and Nutrition in Rheumatoid Arthritis: Can Diet Influence Disease Development and Clinical Manifestations?

**DOI:** 10.3390/nu12051456

**Published:** 2020-05-18

**Authors:** Chiara Gioia, Bruno Lucchino, Maria Grazia Tarsitano, Cristina Iannuccelli, Manuela Di Franco

**Affiliations:** 1Dipartimento di Scienze Cliniche, Internistiche, Anestesiologiche e Cardiovascolari-Reumatologia, Sapienza University of Rome, 00161 Roma, Italy; chiara.gioia@uniroma1.it (C.G.); c.iannuccelli@policlinicoumberto1.it (C.I.); manuela.difranco@uniroma1.it (M.D.F.); 2Dipartimento di Medicina Sperimentale, Sapienza University of Rome, 00161 Roma, Italy; mariagrazia.tarsitano@uniroma1.it

**Keywords:** Rheumatoid Arthritis, diet, nutrition, pathogenesis, Mediterranean Diet

## Abstract

Rheumatoid arthritis (RA) is a systemic, autoimmune disease characterized by joint involvement, with progressive cartilage and bone destruction. Genetic and environmental factors determine RA susceptibility. In recent years, an increasing number of studies suggested that diet has a central role in disease risk and progression. Several nutrients, such as polyunsaturated fatty acids, present anti-inflammatory and antioxidant properties, featuring a protective role for RA development, while others such as red meat and salt have a harmful effect. Gut microbiota alteration and body composition modifications are indirect mechanisms of how diet influences RA onset and progression. Possible protective effects of some dietary patterns and supplements, such as the Mediterranean Diet (MD), vitamin D and probiotics, could be a possible future adjunctive therapy to standard RA treatment. Therefore, a healthy lifestyle and nutrition have to be encouraged in patients with RA.

## 1. Introduction

Rheumatoid arthritis (RA) is a chronic autoimmune and inflammatory disease, characterized by joints involvement as well as systemic features [1]. The global prevalence of the disease is estimated around 1–2%, with a large variation among different populations [2]. Genetic and environmental factors interact during RA pathogenesis, a multistep process that begins years before clinical onset of the disease. The most relevant genetic risk locus associated to RA is found in the HLA class II molecule-encoding locus (chromosomal position 6p21.3). Several HLA-DRB1 alleles, encoding a common amino acid sequence at the position 70–74 in the third region of the DRβ1 chain, the so-called “shared epitope (SE)”, have been associated with an increased risk of seropositive RA [3]. In genetically predisposed individuals, the influence of environmental factors can lead to a breaking of immune tolerance to self-antigens, including citrullinated and carbamylated proteins [4]. Several environmental factors, such as cigarette smoking, air pollution, dust, diet and infections, contribute to the development of systemic autoimmunity and autoantibodies appearance years before the onset of symptoms [5,6,7]. Diet and nutrients have received considerable attention as potential environmental factors influencing the development and the course of the disease. Although a number of studies suggested associations between dietary habits, mainly regarding fruit, vegetables or meat intake and the disease’s development, the results are still inconclusive [8,9,10,11,12,13,14]. In recent years, an increasing number of studies have investigated the role of diet and nutrition as potential tools for RA prevention and management [15]. It can be assumed that the Mediterranean Diet (MD), together with genetic and other lifestyle factors, could explain the lower RA incidence in Southern Europe (0.3–0.7%) compared with Northern Europe and North America (0.5–1.1%) [2]. The aims of this review are to analyze the role of diet and nutrition as factors influencing RA development and the effect of diet on RA disease activity.

## 2. Rheumatoid Arthritis (RA) Pathogenesis: Diet as a Risk Factor

Dietary habits could represent both disease risk and protective factor, based on the properties of specific foods. Specific dietary choices can indeed show pro-inflammatory effects (for example red meat, salt, excessive caloric intake) or on the contrary reduce inflammation (oil, fatty fish, fruit and others) [16,17]. The prevalence distribution of RA shows a higher number of RA patients in Western countries, in opposition with Eastern world and developing countries [18]. The Western diet, characterized by a high intake of red meat, saturated and trans fats, a low ratio of omega-3:omega-6 fatty acids and high consumption of refined carbohydrates, has been associated with an increased RA risk principally through an increase of inflammation and an induction of insulin-resistance and obesity [19,20]. Even if the relationship between RA and diet is not as strong as other risk factors (smoke primarily), dietary influence has been widely studied, also considering the complex nature of nutrient provision. Moreover, diet represents a major factor influencing microbiota composition, which has been involved in the disease’s development (Figure 1).

### 2.1. Diet and Inflammation

RA and other chronic diseases, including cardiovascular disease (CVD), type 2 diabetes mellitus, different types of cancer, and Alzheimer’s disease, share inflammation as a driving pathophysiological process. Indeed, C-reactive protein (CRP), a marker of inflammation, predicts future development of hypertension and diabetes mellitus better than body mass index [21]. The majority of chronic diseases (e.g., obesity, diabetes) are strongly influenced by nutrition, and food metabolism is closely related to inflammatory processes [22]. Postprandial inflammation is a component of the normal stress reaction to food in cells. Nutrients can modulate the inflammatory status of humans and consequently the pro- or anti-inflammatory properties of specific foods and components have emerged in nutrition sciences [23]. Among different nutrients, carbohydrates and fats are the most widely studied. Carbohydrates’ quality (glycemic index) more than absolute carbohydrates’ quantity represents the stronger influence on systemic inflammation: a diet with high intake of fiber, lowering the velocity of carbohydrates absorption, presents a negative relationship with CRP, interleukin-6 (IL-6) and tumor necrosis factor alfa (TNF-α), mediators of inflammation [24]. Regarding fat, several studies reported pro-inflammatory effects of trans-fatty acids, with increased levels of TNF-α, IL-1, CRP and vascular endothelial dysfunction; in contrast, the polyunsaturated fatty acids (PUFA) omega-3, mainly present in fish oils, present inverse correlation with IL-6, matrix metallopeptidases 3 (MMP3) and CRP levels. The Mediterranean Diet (MD) features mainly vegetables, unrefined cereals, fruit, legumes, fish and extra-virgin olive oil, associated with a moderate intake of eggs, poultry, dairy products and low consumption of refined sugar and red meat. Red wine is also included, and herbs and spices are largely used [25,26,27]. This dietary pattern, present mainly in Southern Europe and in olive-growing regions, is opposed to the Western diet. Featuring an omega3:omega 6 ratio of 1:7 [28], related to a high intake of alfa-linolenic acid (ALA), a specific “anti-inflammatory” PUFA, MD is associated with a reduction in total and cardiovascular mortality, and cancer, Alzheimer’s and Parkinson’s disease incidence. Several foods included in MD show direct anti-inflammatory effects: extra-virgin olive oil use is associated with a reduction of thromboxane 2 (TXB2) and leukotriene B4 (LTB4), not observed with corn oil and non-virgin olive oil. The consumption of tomato drink for 26 days induces a reduction in TNF-α production. Black tea consumption, exerting an anti-inflammatory effect mainly for its flavonoid content, can decrease CRP levels, leukocyte and platelets aggregation and activation, reducing inflammatory response in healthy men. Moderate red wine intake contributes to a reduction in low-/high-density lipoprotein (LDL/HDL), oxidized LDL, CRP and fibrinogen and to an increase in HDL and total antioxidant capacity [21]. Epidemiological studies also suggested anti-inflammatory properties of flavonoids and carotenoids, groups of natural substances found in fruits and vegetables widely included in MD. In particular, flavonoids are able to inhibit both isoforms of inducible nitric oxide synthase (iNOS) and of cyclooxygenase (COX-2), which are involved in the production of inflammatory mediators; they also downregulate adhesion molecules and increase antioxidant defenses, contributing to enhance their anti-inflammatory capacity [29]. Tomato, largely included in MD, is an important source of lycopene (a carotenoid compound), one of the most potent antioxidants [30]. Lycopene provision from tomato and tomato products, exerts beneficial effects on cardiovascular (CV) risk, reducing LDL-cholesterol and improving endothelial function, as well as on inflammation, reducing inflammatory factors (CRP, IL-6) and adhesion molecules (ICAM-1) expression [31]. Potatoes, included as starchy food in MD, provide key nutrients to diet, including vitamin C, potassium and dietary fiber. Moreover, several potato components contribute to lower blood pressure, to improve lipid profile and to decrease markers of inflammation (CRP, IL6) [32].

### 2.2. Dietary Habits and the Risk of RA Development

Excessive consumption of red meat and a high total protein supply have been associated with an increased risk of inflammatory polyarthritis [10]. The possible explanations of this association lie in the increased inflammation derived by meat fats and nitrites, as well as an increased synovial involvement secondary to the excessive oral iron load [14,33]. However, a multi-center Chinese case-control study, monitoring the dietary pattern of 968 RA patients and 1,037 healthy controls for 5 years before the clinical onset of the disease, showed no significant differences in red meat consumption between the two groups. RA patients presented a lower intake of fish, potatoes, mushrooms and organ meats; mushrooms, citrus fruits and dairy products consumption showed a protective effect on RA, while potatoes and other fruits’ consumption was associated with an increased risk [34]. Similar results on red meat have been reported by Benito-Garcia et al., who prospectively assessed risk for RA in relation to the consumption of protein, iron and meat among women in the Nurses’ Health Study (NHS). There were no associations between protein, iron, and corresponding food sources with RA risk [12]. Table 1 shows different populations, dietary patterns and RA outcomes of these studies.

A high dietary sodium (salt) intake, common in Western countries, has been associated with an increased risk of RA [35]. High levels of salt could potentiate the detrimental effect of other environmental factors, smoking in particular, inducing serum glucocorticoid kinase-1 (SGK-1) expression, with an increased Th17 lymphocyte differentiation and enhanced autoimmunity [36]. High sodium intake in smokers increases indeed the risk of anti-citrullinated protein antibodies (ACPA) positivity [37]. Accordingly, smoking and salt may have a mutual enhancement effect, but larger studies are needed for more defined results. 

To date, data about milk and dairy products are controversial. The EPI(EPIC-Norfolk) study [10] reported a positive but not statistically significant association between milk consumption and RA development while the Iowa Women’s Health Study observed an inverse association between milk products intake and RA risk [38]. Recently, a large prospective Swedish cohort study reported no association between dairy products’ consumption and RA development [39]. It has been hypothesized that milk could be associated with RA development triggering an allergic reaction to cow milk proteins [40,41], although only a small proportion of RA patients investigated for mucosal sensitivity presented signs of milk proteins reactivity [42]. On the contrary, a protective effect of milk could be related to its vitamin D content [38].

Red meat, eggs and dairy products are the main sources of trimethylamine-N-oxide (TMAO), a pro-inflammatory metabolite deriving from choline and carnitine metabolism [43,44]. Choline and related metabolites, including TMAO, mainly deriving from diet, have been associated with cardiovascular inflammation. These metabolites have been identified in blood samples, synovial fluid and tissue of mice models of arthritis as well as in human studies on CV risk, but their role in RA has not been fully investigated [45].

Omega-3 fatty acids appear to be protective for RA [46]. Different studies analyzed fatty acids as possible protective nutrients for RA. Di Giuseppe et al. observed a reduction (35%) of RA risk in women with a dietary intake of long-chain n-3 polyunsaturated fatty acids higher than 0.21 g/day [47]. A Danish prospective population-base cohort study confirmed a decreased risk of RA (about 49%) for patients with dietary intake of 30 g fatty fish (>8 g fat/100 g fish), while a medium consumption of fat fish (3–7 g fat/100 g fish) was associated with a significantly higher risk of RA [8]. On the contrary, the Epidemiological Investigation of Rheumatoid Arthritis (EIRA) study, a population-based case-control study evaluating more than 4000 participants, rebutted these results, demonstrating only a mild decrease in RA risk with regular consumption of fatty fish [48]. The risk reduction of RA associated to a higher intake of omega-3 fatty acids is related to their anti-inflammatory properties, inhibiting leucocyte chemotaxis, adhesion molecule expression and production of pro-inflammatory leukotrienes and prostaglandins from omega-6 fatty acids [49]. 

The EPIC-Norfolk study showed a reduced intake of fruits and vitamin C in patients with inflammatory arthritis compared with controls, and a decreased risk of RA development in olive oil consumers [11,50]. Fruits, vegetables and olive oil could decrease RA risk, providing several nutrients with antioxidant properties, such as tocopherols contained in olive oil acting as free radical scavengers [51]. However, the already mentioned Danish study demonstrated no association between RA risk and the intake of citrus fruits, vegetables, retinol, beta-carotene, vitamins A, E, C, D, zinc selenium, iron and meat [47]. Although vegetables are rich in antioxidants and provide generally anti-inflammatory effects, some of them such as potatoes, tomatoes and eggplants have been associated with an increased RA risk. Recently it has been shown that solanine, a glycoalkaloid contained in these foods, can increase intestinal permeability with a potential harmful effect on RA development [52].

High sugary drinks consumption has been associated with RA development. The NHS [53] demonstrated that regular consumption of sugar-sweetened soda increases the risk of RA; high-fructose sweetened soft drinks may promote arthritis development in young adults, probably inducing an excessive accumulation of glycation products that support inflammation [54].

Mikuls et al. found an increased risk of seropositive RA in consumers of 4 or more cups of coffee/day, while consumers of more than 3 cups of tea per day presented a reduced risk [55]. Heliovaara et al. documented a positive association between coffee consumption and rheumatoid factor (RF) positivity [56]. However, when cigarette smoking as a confounding factor was included in the analysis, the significance of association was lost. Indeed, increased coffee consumption has been related to a smoking habit [57]. Additional data of a meta-analysis concluded that RA development could be associated with total coffee intake but not with caffeinated or decaffeinated coffee and neither to total caffeine intake. The specific coffee preparation technique was not investigated in these studies [58]. On the contrary, specific molecules and products used in coffee processing and preparation may have a role in increasing RA risk: diterpene cafestol, found in unfiltered coffee brews, could increase RF and the risk of RA [56,59]. Moreover, solvent used for extraction of caffeine from beans have been related to RA risk, as well as other connective tissue diseases [60]. 

The overall quality of the dietary pattern, including the combination of various beneficial or harmful foods, can also have an impact on the risk of RA development. A multivariate analysis of women enrolled in NHS and NHS II showed that women with a healthier overall dietary pattern, measured through the calculation of the dietary quality score 2010 Alternative Healthy Eating Index (AHEI-2010), had a significant reduction of the risk of RA development compared with women with poorer dietary habits. In particular, women with the highest AHEI-2010 quartile, aged 55 or less, showed a 33% RA risk reduction compared to women with the lowest AHEI-2010 quartile. Considering that several foods and nutrients included in the AHEI-2010 score calculation have been singularly linked to the risk of RA, the results of this study suggest that the combination of various foods in a globally healthy dietary style may be more beneficial than the individual food or nutrients on the risk of the disease development [61].

### 2.3. Changes in Body Composition and the Risk of RA

Body composition can influence the risk of RA development. In particular, obesity, increased body mass index (BMI) and waist circumference are risk factors for RA [62,63]. Obesity is characterized by excessive fat accumulation caused by a dietetic caloric surplus, and is associated with several health issues and systemic inflammation [15]. White adipose tissue could be considered an “endocrine organ”, able to release pro-inflammatory mediators such as interferon-alfa (INF-α), IL-6, CRP and adipokines [64]. The increased production of pro-inflammatory molecules creates a favorable environment for autoimmunity development. Leptin is a pro-inflammatory adipokine that promotes inflammatory cytokine production by macrophages, granulocyte chemotaxis, dendritic cell maturation and Th1 and Th17 responses, and inhibits regulatory T-cell functions [65]. Leptin inhibits B-cell apoptosis and promotes the survival and proliferation of autoreactive cells in the preclinical phase of RA [66]. Adiponectin is an adipokine with pleiotropic effects and shares structural characteristics with TNF-α. In RA, adiponectin contributes to the development of an inflammatory environment in synovia and promotes osteoclastogenesis [67,68,69]. Visceral obesity and the metabolic syndrome (MetS), characterized by hypertension, dyslipidemia and insulin resistance, increase inflammatory diseases development risk [34]. MetS is often present in early-diagnosed RA and is associated with an increased risk of RA itself [70]. Recently, a large population-based study, investigating more than 500,000 subjects, suggested that an increased waist circumference is associated with an increased risk of RA development even after correction for BMI [71]. In established disease, up to two-thirds of RA patients present a phenotype characterized by muscle wasting and fat mass gain, without body weight variations, termed at its extreme “sarcopenic obesity”, which is more evident in women [72]. Pro-inflammatory cytokines are active on striate muscles, promoting proteolysis of sarcoprotein and determining the loss of muscle mass [73]. However, lifestyle factors, such as diet and physical inactivity, as well as glucocorticoids treatment are important factors involved in sarcopenic obesity development among patients with RA [74,75,76]. Several studies in early RA (ERA) patients revealed that lean mass loss and fat mass gain can be evident also in the early stage of the disease, although the involved factors are currently unknown [76,77]. Body composition in treatment-naive, ERA patients has been evaluated in a study by Turk et al. Loss of muscle mass was 4–5 times more common in ERA patients than controls, although there were no associations between disease activity and an unfavorable body composition [78]. In a large, population-based, Danish cohort, a higher total body fat and a higher waist circumference were directly associated with an increased RA development risk among women, suggesting that the unbalance between the fat and lean tissues occurs early in the course of the disease and may have a role in its pathogenesis [79].

### 2.4. Diet, Microbiota and RA Risk 

The gut microbiota, including all the commensal and potentially pathogenic bacteria residing in the human gastrointestinal system, plays a major role in the physiologic and immunologic homeostasis of the organism and its alteration could be related to the pathogenesis of several inflammatory diseases, including RA. The majority of gut bacteria belong to the Bacteroides and Firmicutes phyla, although the presence of other phyla, such as Proteobacteria, Fusobacteria and Actinobacteria compose and support the homeostasis of intestinal microflora [80,81]. They have important digestive roles: vitamin synthesis, digestion and cleavage of fiber and other dietary components into metabolites. The principal final product of fiber catabolism by the gut microbiota is short-chain fatty acids (SCFA), which present anti-inflammatory effects modulating macrophages and dendritic cells and enhancing regulatory T cells function [82,83,84,85]. The link between the gastrointestinal microbiota and the immune system has been largely studied; an altered epithelial and mucosal permeability influences the immune tolerance to local microbiota and can unbalance immune system towards inflammatory reactions [86,87,88]. In mouse models, an enrichment of segmented filamentous bacteria in the gut can promote the differentiation of Th17 cells in lamina propria and trigger the development of inflammatory arthritis [89,90]. Changes in microbiota or dysbiosis, influenced also by lifestyle and dietary patterns, may consequently promote intestinal increased permeability and local inflammation, causing a consequent spreading of inflammation to the joints [83]. Significant modifications of the intestinal microbiota occur in RA. RA patients present a reduced gut microbial diversity in comparison to healthy controls [86,91,92] as well as qualitative modifications. Specific bacteria genera/species could directly contribute to the development of the disease. *Colinsella* sp., a bacterial species enriched in RA microbiota, can trigger inflammation increasing gut permeability and IL-17A production [93,94,95]. By contrast, the reduced *Faecalibacterium* abundance in RA microbiota can be associated to a reduced production of butyrate, the final metabolite of fiber cleaving, which presents an anti-inflammatory effect, preserving mucosa integrity [80,96]. Several authors reported an abundance of *Prevotella copri* (*P. copri*) in early RA patients [91,97] with a significant effect on the inflammatory response: an increase of related Th17 cytokines (i.e., IL-6, IL-23), an increased gut permeability with a major penetration of external antigens and bacteria, and a mutual decrease of protective species [98]. *P. copri* 16S RNA levels on stool samples correlated also with disease activity in new-onset untreated RA [91]. *P. copri* is able also to metabolize choline in to TMAO, a pro-inflammatory metabolite already mentioned. The pre-clinical and early phases of RA are characterized also by a reduced presence of *Bifidobacterium* and *Bacteroides* bacterial genera [93,96,99,100]. Both *Prevotella* and Bacteroidetes are diet-responsive bacteria: in particular, the increase in *Prevotella* is related to high-fiber diet, while an increase in Bacteroidetes to a diet rich in fat and animal protein [97,101]. Western diet, rich in animal protein and saturated fats, enhances *Bacteroides* enterotype; in contrast *Prevotella*-driven enterotype is found in carbohydrate-based diet (simple sugars, fiber) [97,102]. Dietary patterns, acting through a possible reversal of the gut dysbiosis, have been hypothesized as a possible strategy to ameliorate or even prevent autoimmune diseases [103]. Opposite to the Western diet, MD is considered one of the healthiest dietary patterns today. Foods rich in fibers, present in MD, are degraded by Firmicutes and *Bacteroides* into SCFA [104,105]. Fiber intake could have positive impact on microbiota and SCFA, derived by fiber cleaving, have beneficial effects on intestinal barrier [95]. Butyrate, a SCFA, increases tight junctions (TJ) protein expression reducing intestinal permeability and bacterial translocation, limiting local and systemic inflammation. Accordingly, a possible beneficial effect of MD in RA, acting through changes in microbiota composition, has been suggested [105]. Despite a study failing to demonstrate that two months of MD adherence in RA and fibromyalgia patients were associated to any significant change in gut microbiota composition, recent studies suggest that a longer period (more than 3 months) of MD is needed to produce a significant increase in the diversity of microbiota [106,107]. Accordingly, additional research is necessary to elucidate the links between the health-promoting effect of the MD on gut microbiota and the possible effect in preventing RA development [104].

## 3. Protective Dietary Styles and Nutrients in Subjects at Risk for RA

### 3.1. Mediterranean Diet (MD)

The health benefits of the Mediterranean Diet (MD) have been largely studied in recent years [26,27]. Strong evidence, based on a meta-analysis including more than 12,800,000 subjects adhering to MD, supports the decreased risk of overall mortality, cardiovascular disease (CVD), overall cancer incidence, neuro-degenerative disease and diabetes [108,109]. MD’s protective properties are related to antioxidant and anti-inflammatory effect of peculiar abundance of several nutrients, especially mono-unsaturated fatty acids, polyphenols, tocopherols, that promote a reduction in insulin-resistance and protect from diabetes and CVD [16,26,109,110]. The anti-inflammatory properties of MD may potentially halt RA development and progression. However, a limited number of studies is currently available in literature, with mostly inconclusive results. Hu et al. analyzed the potential protective effect of MD on the risk of RA development in women enrolled in the NHS I e NHS II. No significant protective effect was highlighted [111]. Similarly, a Swedish nested case-control study, conducted by Sundstrom et al., found no protective effect of MD followed for a median period of time before the onset of symptoms of 7.7 years on the risk of RA development [112]. Forsyth et al. published a recent systematic review investigating the potential efficacy of MD in preventing RA development and improving the clinical manifestations. Despite of a limited number of studies included, the analysis found no substantial reduction in the risk of RA with MD, although MD nutritional style could provide a symptomatic improvement, including disease activity, inflammatory markers and physical function [113].

### 3.2. Vitamin D

Vitamin D has been extensively studied in the past decade [114], after the demonstration of a link between its deficiency and several health issues. 1,25-dihydroxyvitamin D (1,25(OH)2D), the active metabolite of vitamin D, shows potent anti-proliferative, antibacterial and anti-inflammatory properties in vitro [115]. The immunomodulatory effects of 1,25(OH)2D are mediated by the vitamin D receptor (VDR), expressed by several immune cells. The activation of VDR modulates T cell phenotype, suppressing pro-inflammatory Th1 and Th17 and promoting tolerogenic, regulatory T cells. Antigen-presenting cells express the 1α-hydroxylase, an enzyme that converts and activates the precursor 25-hydroxyvitamin D3 25-(OH)D3 to 1,25-(OH)2D3; as a consequence, immune cells are able to both activate and respond to vitamin D [116]. Disrupting this autocrine signal, vitamin D deficiency could interfere with immune system antibacterial and anti-inflammatory responses. Accordingly, vitamin D deficiency may have a role in the progression from preclinical to clinical RA. In transgenic mouse models of RA, the deletion of the VDR gene exacerbated inflammation and was associated to bone loss [117]. Vitamin D deficiency has been observed in many patients with autoimmune disease, RA included, but how and if low serum 25(OH)D contributes to RA risk is less known. Similarly, it is currently not known if the correction of the vitamin D deficiency through supplementation could halt the progression to RA in at risk individuals. A retrospective study, performed on RA patients who had donated blood up to 5 years before the onset of the disease, showed no differences in vitamin D status between patients and healthy controls and no association between the vitamin D deficiency and the subsequent development of RA [118]. Another study investigated the prevalence of vitamin D deficiency based on the serum ACPA status, showing no difference between healthy ACPA-positive subjects and ACPA-negative controls [119]. Regarding the potential prevention of RA development, the Women’s Health Initiative randomized clinical trial, evaluating the effect of calcium plus vitamin D supplementation, demonstrated no benefit on the risk of the disease development, showing the opposite, an increased risk of RA for higher levels of vitamin D intake [120]. However, a meta-analysis conducted on more than 200,000 subjects reported an association between low vitamin D intake and RA risk [121]. Considering clinical and biochemical differences among patients with RA, the role of vitamin D in the pathogenesis of disease can vary on genetic base. Indeed, four different synovial tissue gene expression profiles in RA patients were identified by Dennis et al. [122], probably related also to patients’ disease duration or treatment regimen. It is important to consider difficulties and limitations linked to vitamin D studies. Vitamin D sources are food and sunlight exposure that is subjected to seasonal variations; accordingly, the exact quantification of overall vitamin D “intake” through both food and sunlight exposure is difficult [123]. Another limitation is the definition of vitamin D deficiency itself that leads to the use of different cut-off values across various studies. For example, in the UK the National Institute of Clinical Excellence (NICE) defines vitamin D deficiency when its serum level is <30 ng/mL and insufficiency for levels of 30–50 ng/mL [124]. Other cut-offs described are 20 ng/mL for deficiency and 20–30 ng/mL for insufficiency [125]. To date, the majority of studies analyzed serum levels of the main circulating form of vitamin D, 25-OHD3; other metabolites both in serum and in synovial fluid are gaining interest. Indeed synovial fluid levels of vitamin D metabolites have a stronger relationship with RA disease activity when compared with circulating serum metabolites [126]. In spite of all the limitations in the studies conducted so far, vitamin D anti-inflammatory and beneficial properties need to be better investigated to define their potential role in RA prevention.

### 3.3. Alcohol and Red Wine

Alcohol has an interesting role as a “preventing” RA factor and modulator. Lu et al. found a 30% decreased risk of RA developing in subjects with consumption of 3–5 standard drinks/week compared to no alcoholic drinks [127]. In the Iowa Women’s Health study cohort, no associations were identified between alcohol intake and the risk of RA; it is important to underline that this cohort presented an overall lower consumption of alcohol than studies showing associations [128]. Furthermore, a Swedish case-control study on 386 patients with pre-clinical RA did not report associations between RA and low alcohol consumption [112]. Despite the observation of a protective effect of moderate alcohol intake on the development of different chronic diseases, including RA, a definitive indication cannot be made, considering several limitations of the available studies, among which confounding factors and classification issues; accordingly, the results of these studies must be interpreted with caution [129,130,131]. A different consideration must be taken for red wine, which has a large content in polyphenols that can reduce inflammation, improve lipid metabolism, antioxidant state, and endothelial function independently from the alcohol content [132].

### 3.4. Green Tea

Epigallocatechin-3-gallate (EGCG), the main phytochemical present in green tea, has protective effects, demonstrated on cardiovascular, inflammatory, and neurodegenerative disease, as well as different types of cancer [133]. In RA, EGCG is able to downregulate Mcl-1, an anti-apoptotic protein, in synovial fibroblast, consequently facilitating these cells’ apoptosis [134]. Moreover, ECGC inhibits IL-1B induced IL-6 production by synovial fibroblast and avoids bone and cartilage destruction through a downregulation of MMP-1, MMP-2 and MMP-3 in synovial fibroblasts [135]. Although EGCG showed the capacity to suppress inflammation in mouse models of arthritis, more studies are needed to investigate its therapeutic potential in humans [136].

## 4. Effects of Diet and Nutrients on Disease Activity in Established RA 

In the context of personalized medicine, patients would have a more active role in their own disease management. Apart from pharmacological treatment, patients are often interested in self-management strategies for symptomatic improvement, such as resting, applying heat or cold, attending physical therapy, and/or using a splint or brace [137]. A suggested dietary habit, which foods are recommended, and which should be avoided, are some of the emergent questions that patients address to rheumatologists. Diet could modulate and enhance pharmacological treatment efficacy and consequently, is gaining interest among rheumatologists. In recent years, possible mechanisms for dietary intake as possible adjunctive therapy for RA have been studied [138]. Even if detailed studies in humans on the relationship between dietary intake and immune and inflammatory pathways in RA are lacking, different potential mechanisms have been proposed based on in vitro or in animal models, especially for omega-3 and omega-6 fatty acids.

### 4.1. Omega 3 and Omega 6

Omega-3 and omega-6 fatty acids are important component of phospholipid membranes and can modulate inflammation mediators, once metabolized to eicosanoids [139,140]. Deriving primarily from fatty fish but also poultry, nuts and berries [141], omega-3 fatty acids (eicosapentaenoic acid (EPA) and docosahexaenoic acid (DHA) are lipid mediators involved in the limitation and in the resolution of inflammatory responses [142]. Arachidonic acid (AA) and omega-6 fatty acids derived from animal food sources (meat, eggs, dairy products) have, on the contrary, mainly a pro-inflammatory role. Indeed, AA is the precursor of lipid mediators of inflammation, namely prostaglandins and leukotrienes, the main actors together with neutrophils in the acute inflammatory response. LTB4 and prostaglandin E2 (PGE2), deriving from AA metabolism, play a central role as chemoattractant for inflammatory cells and as mediator of vasodilatation and increased vascular permeability, responsible of edema and exudate formation [143,144]. Linoleic acid (LA) (omega-6 precursor) and ALA (omega-3 precursor) are the only essential fatty acids that the body cannot synthesize. The main dietary sources of LA include plant oils but also cereals, animal fats and wholegrain bread; ALA derives from green leafy vegetables, flaxseed and rapeseed oils [145]. Delta-5 and -6 desaturases are the enzymes responsible for production of DHA and EPA from LA, as well as of di-homo-gamma linolenic acid (DGLA) and AA from ALA. The affinity of these enzymes for ALA is higher than for LA [146]. The optimal omega-6:omega-3 ratio from dietary intakes should be 1–4:1, but in the Western diet, where LA is more abundant than ALA, this ratio is increased to 10–20:1. This unbalanced ratio, typical of the Western diet, can shift the action of desaturases towards LA, with a greater production of pro-inflammatory mediators [147]. Several studies investigated the immunological effects of omega-3 fatty acids. In vitro, EPA and DHA reduce the production of pro-inflammatory eicosanoids by human macrophages [140] and can reduce neutrophil chemotaxis and adherence to endothelium induced by LTB4 [148]. Omega-3 fatty acids also have an in vitro inhibitor effect on NOD-like receptor NLRP3 inflammasome in human macrophage [149]. Moreover, omega-3 fatty acids downregulate adhesion molecules on vascular endothelial cells (VCAM-1, ICAM-1, E-selectin) with consequent decreased inflammation [150,151,152]. EPA and DHA promote inflammation resolution, stimulating macrophage phagocytosis and reducing local neutrophil invasion. EPA and DHA are indeed precursors of local lipid mediators of inflammation resolution, defined specialized pro-resolving mediators (SPM), including resolvins, protectins and maresins; they reduce neutrophils infiltration and enhance macrophage activity on apoptotic cells, debris and microbes [153]. Since the 1990s, several trials of fish oil supplementation have been performed in RA (Table 2). Most of them show an anti-inflammatory effect of fish oil supplementation in RA; meanwhile, the effects on RA disease of omega-3 derived from diet were not evaluated. 

### 4.2. Caloric Restriction

In RA, an altered inflammatory response and an enhanced activation of NLRP3 inflammasome stimulate interleukin-1 beta (IL-1B) production, contributing to inflammatory manifestations [153]. A ketogenic diet, with a low carbohydrate dose, or a caloric restriction showed anti-inflammatory effect, with a reduced NLRP3 inflammasome-mediated IL-1β production [162,163,164]. In a murine model, beta-hydroxybutyrate (BHB), a ketone body produced in starvation state, inhibits NLRP3 inflammasome activation in macrophages, and down-regulates IL-1B production by human monocytes [165]. Two studies investigated the anti-inflammatory effect of a caloric restriction in obese and non-obese subjects. The first prospective study showed a reduced expression of NLRP3 inflammasome and IL-1 B in adipose tissue [166], while the second documented lower serum levels of TNF alfa in caloric restriction group compared to the control arm [167]. Therefore, caloric restriction could have potential beneficial effects on RA symptoms and disease activity by inhibiting the NLRP3 inflammasome.

### 4.3. Antioxidant

Antioxidant nutrients, variably present in different foods, have the main function to act as scavengers of free radicals and present several important biological effects. They inhibit tumor cells proliferation and cholesterol absorption, present an anti-inflammatory effect, and modulate many redox reactions [168]. Among these functions, prevention and delay of atherosclerosis is one of the most relevant: epidemiological, in vivo and in vitro studies suggesting different preventive effects in atherogenesis, including the reduction of low-density lipoproteins oxidation, oxidative stress, chemotaxis, inflammation, release of nitric oxide, cell adhesion, and platelet aggregation [169,170]. Rheumatoid arthritis features oxidative stress, increased formation of reactive oxygen species, lipid peroxidation, DNA damage and a decreased activity of antioxidant protective systems [171]. In a study conducted on 40 RA female patients, daily supplementation with antioxidant (50 μg selenium, 8 mg zinc, 400 μg vitamin A, 125 mg vitamin C, and 40 mg vitamin E) ameliorated oxidative stress and disease activity but not the number of tender and swollen joints [172]. By contrast, in another study, the supplementation of vitamin A, C, selenium and zinc did not improve RA disease activity [173]. Sahebari M et al. reported no correlation between disease activity and low serum concentrations of zinc and selenium [174]. Silicon, present in solid plant foods as cereals and certain vegetables, is involved in redox balance and inflammation. A study evaluating the relationship between plasma silicon levels, redox status (total antioxidant status) and inflammatory markers (CRP, IL-6) in RA, reported that patients affected by RA presented elevated plasma silicon levels compared to controls. No correlation between plasma silicon and dietary silicon level was observed, although in RA patients a negative correlation between silicon intake and IL-6 and total oxidant status was found [175]. In a recent randomized, double-blind, placebo-controlled clinical trial, active RA patients received 1500 mg ginger powder (or placebo) daily for 12 weeks. The ginger-treated group presented a reduction of disease activity, associated to increased *Fox-P3* genes expression and decreased *ROR-yt* genes, suggesting a potential modulation of T cells phenotypes by ginger [176].

### 4.4. Flavonoids

Flavonoids, phenolic compounds present in plants and fungi, have antioxidant, antimicrobial and anti-inflammatory properties. Genistein, the major active compound found in soybean, features anti-inflammatory, anti-angiogenesis, immunomodulatory, analgesic and chondroprotective properties. In vitro and in vivo studies showed the efficacy of genistein to inhibit IL-1B, TNF-α, EGF-induced proliferation and MMP-9 expression in fibroblast-like synoviocytes of RA [177,178].

### 4.5. Gluten

Gluten, a complex mixture of hundreds of distinct proteins present in wheat grains, mainly gliadin and glutenin, triggers immunological response mainly in coeliac disease. Recent studies suggest that gluten may represent an antigen also in RA, driving an altered immune response [179]. In RA patients, a study showed that a gluten-free, vegan diet followed for 1 year was associated with a significant decrease of anti-beta-lactoglobulin and anti-gliadin antibodies levels, as well as a decrease of disease activity [180]. In another randomized study on 66 RA patients, the same dietary pattern also showed athero-protective and anti-inflammatory properties, observing LDL and oxidized LDL levels reduction [181]. 

### 4.6. Fasting

Subtotal fasting is characterized by a limited intake of carbohydrate and energy through vegetable juice and vitamin and mineral supplementation. Reducing the number and the activation of CD4+ lymphocytes, fasting may promote a reversal of the typical RA immunological status, characterized by activation of CD4+ T cell and differentiation to Th1 and Th17 lineages. A 7–10 days period of fasting could create a transient immunosuppression, reducing T cell activation [182]. Although a reduction of inflammation (ESR, CRP) and pain following fasting have been demonstrated, these are transient with no long-term modifications on disease activity [163,183].

### 4.7. Vegan Diet

A vegan diet, providing antioxidants, lactobacilli and fibers, improves intestinal flora composition, with benefits on RA disease activity [184]. In a single-blind dietary intervention study, 24 patients with moderate to severe RA followed a very low-fat (< about 10%) vegan diet for 4 weeks. After this period, weight and RA symptoms, except duration of morning stiffness, decreased significantly [185]. 

### 4.8. Vitamin D

Several data showed that hypovitaminosis D status is associated with RA disease progression; it is necessary to consider that disease severity can also restrict patient mobility, limiting access to ultraviolet (UV) light and thus diminishing conventional epidermal synthesis of vitamin D. Several epidemiological studies reported an inverse association between serum 25(OH)D concentrations and RA disease activity and severity [186], even early in the course of the disease [187]. Recently, a meta-analysis identified 9 randomised control trials (RCTs) of vitamin D supplementation in rheumatic diseases, including 5 studies of RA patients [188]. Among patients with RA, the vitamin D supplementation group showed a decreased rate of disease flares, pain levels and Disease Activity Score 28 (DAS-28), not reaching however, statistical significance. Similar findings have also been reported in another meta-analyses on RA patients [189]. Conversely, another meta-analysis of 11 studies showed that low vitamin D intake was linked with both increased risk of RA and greater disease activity [121]; Nevertheless, the latter included only cohort studies and not RCTs, limiting definitive conclusions. 

### 4.9. Alcohol

Available data regarding the effect of alcohol on RA disease activity are controversial. In a healthy population, a moderate overall alcoholic intake has been associated with lower levels of inflammatory cytokines [190,191,192]. In the Västerbotten Intervention Program (VIP) cohort, investigated by Sundstrom et al., alcohol intake was not associated with risk of RA [112]. On the contrary, in a prospective, observational study of 615 RA patients, alcohol showed harmful effects on CRP, DAS28, visual analogue scale (VAS) pain, modified health assessment questionnaire (HAQ) and radiographic damage [193]. In a Swedish prospective study, early RA patients (<2 years), who were drinkers, presented a very small reduction in self-reported tender joint count, patient global, pain VAS and HAQ compared to non-drinkers [194]. However, the potential benefits of alcohol use must be evaluated in sight of the potential harm deriving from pharmacologic interactions, especially considering that the combination of alcohol with methotrexate may enhance hepatotoxicity danger.

### 4.10. Probiotics

Probiotics are defined by the Food and Drug Administration (FDA) as “live microorganisms which, when administered in adequate amounts, confer a health benefit on the host” [195]. Probiotics consumption seems to reduce oxidative stress in human body [196]. *Lactobacillus* and *Bifidobacterium* are the main probiotics used in commercial and pharmaceutical products [197]. Based on microbiome role in RA pathogenesis and its changes in RA patients, probiotics could represent an adjunctive therapy in RA. A meta-analysis of 9 studies reported a reduction of the pro-inflammatory cytokines IL-6 levels but not of disease activity score [198]. In a randomized double-blind clinical trial, 60 established-RA female patients were randomized to receive *Lactobacillus casei* 01 or a placebo for 8 weeks. At the end of the study, the intervention group presented improvement of symptoms (tender and swollen joint count, global health score, DAS28) and different levels of IL-10, IL-12, TNF-α, in favor of the probiotic group [199]. Other studies, focusing mainly on metabolic changes associated with probiotics use, did not observe improvement in serum lipids or in oxidative status in RA patients after daily supplementation of *Lactobacillus casei 01* [200,201]. A recent randomized, single-blinded, controlled crossover trial, ADIRA (Anti-Inflammatory Diet in Rheumatoid Arthritis), with the aim to investigate the potential adjunctive advantage on disease activity of an anti-inflammatory dietary pattern, rich in n-3 fatty acids, dietary fibers, and probiotics, compared with a control diet nutritionally similar to a typical Swedish diet high in saturated fatty acids (SFAs), has been reported. The main meals in diet contained fish (prevalently salmon) 3-4 times/week and vegetarian dishes with legumes 1–2 times/week. Potatoes, whole-grain cereals, vegetables, yogurt for sauces, spices and other flavoring were included. The probiotic used was *Lactobacillus plantarum* 299c, 5 days/week. Patients answered lifestyle and food questionnaires. Even if a mild symptomatic improvement during ADIRA intervention period was observed, no significant difference in effects on DAS28 and its components between the ADIRA diet and Swedish diet was found [202].

## 5. Dietary Recommendation for RA Patients

Considering all the evidence reviewed, healthy dietary habits may represent a useful tool in reducing the risk of RA, related comorbidities, and RA progression and disease activity. Mediterranean diet is the most encouraged dietary pattern, coupled with a high consumption of ‘fatty’ fish (sardines, salmon, seabeam, seabass and trout) for their well-known anti-inflammatory properties. Figure 2 summarizes the nutrients and their dietary sources that are involved in RA risk development and progression. According to Philippou et al., red meat intake should be limited (1-2/month), extra-virgin olive oil consumption should be daily, along with 1–2/week consumption of fatty fish, weekly consumption of other types of fish and poultry, and high consumption of wholegrains, legumes, 5 or more fruit and vegetables per day, preferably seasonal and locally produced. Sugar-sweetened drinks, salt, alcohol and coffee should be avoided or only moderately consumed. Physical activity and a healthy lifestyle should be combined with dietary patterns, to reach an optimal body weight. Sugar and salt intake should be reduced especially in patients in glucocorticoid treatment. Vitamin D supplementation is important in established RA for bone health, for its anti-inflammatory proprieties and potentially for its benefic effect on disease activity [15]. On the contrary, solid evidence of vitamin D utility in RA prevention is still lacking. In this context, preclinical RA phase and inflammatory arthralgia should be a matter of future research for dietitians, with the aim of altering or slowing down RA development and onset. Body weight control is a primary target, because of its negative impact on disease activity remission and the efficacy of treatment [203,204].

## 6. Conclusions

The evidence of diet’s impact on RA disease activity, together with the role of microbiota in the disease pathogenesis and the beneficial effects of nutrients on inflammation and immunity, underline the importance of defining the best nutritional lifestyle in RA patients. Pharmacological therapy with conventional, biological and small-target disease-modifying anti-rheumatic drugs (DMARDs) could be enhanced by adding a complementary therapy, based on lifestyle changes. Physical activity, weight loss and healthy dietary patterns may represent helpful tools for disease management, promoting a reduction of inflammation, symptoms and disability. Moreover, early manifestations of RA could be potentially delayed with dietary interventions, based on the beneficial effect of vegetarian or vegan diets, and poly-unsaturated fatty acid/oleic acid. The long-term effects of these dietary manipulations could help in reducing RA disease activity, delaying disease progression and probably decreasing the dose of drugs used for treatment of RA patients, globally improving RA patients’ prognoses.

## Figures and Tables

**Figure 1 nutrients-12-01456-f001:**
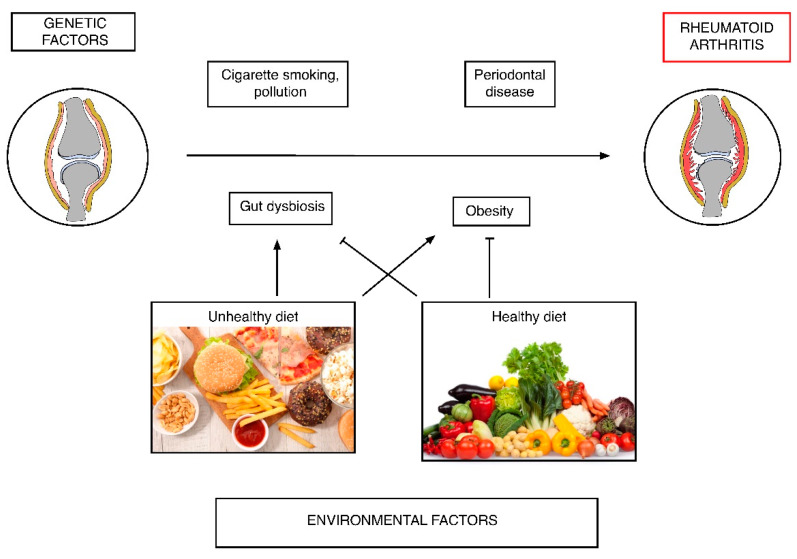
Role of diet in rheumatoid arthritis (RA) pathogenesis.

**Figure 2 nutrients-12-01456-f002:**
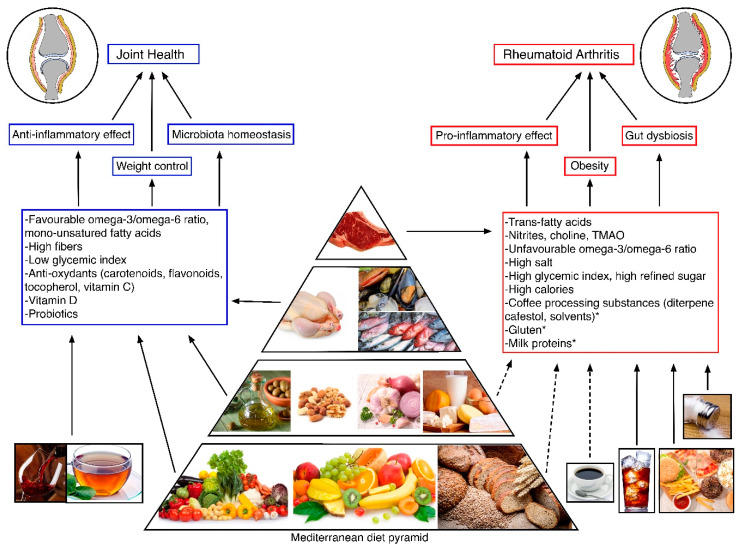
Nutrients and their food sources involved in the development and progression of Rheumatoid Arthritis. * Nutrients with less defined evidence. TMAO: trimethylamine-N-oxide.

**Table 1 nutrients-12-01456-t001:** Studies on effect of different nutrients and their food sources on RA risk.

Study	Duration	Population	Evaluation Method	Nutrients/Foods	Outcomes (Effect on RA Risk)
Pattison et al. (2004) [10]	9 years (1993–2002)	25630 subjects (EPIC-Norfolk) (88 with IP and 1:2 ratio with healthy controls)	Prospective 7-day food diary	1. Red meat 2. Protein intake 3. Vitamin C	1. Harmful effect 2. Harmful effect 3. Protective effect
He J et al. (2016) [34]	1 year (2012–2013)	968 RA patients and 1037 healthy controls	Self-administrated weakly retrospective FFQ (over the last 5 years before RA onset)	1. Potatoes 2. Fruits (no citrus fruits) 3. Citrus fruits 4. Mushrooms 5. Dairy products 6. Red meats 7. Vegetables	1. Harmful effect 2. No effects 3. Protective effect 4. Protective effect 5. Protective effect 6. No effects 7. No effects
Benito-Garcia et al. (2007) [12]	22 years (1980–2002)	**82**,064 women in NHS (546 RA)	Prospective semi-quantitative FFQ at baseline and every 2 years in follow-up	1. Proteins 2. Iron 3. Red meats 4. Fish	1. No effects 2. No effects 3. No effects 4. No effects

EPIC-Norfolk: European Prospective Investigation of Cancer in Norfolk; IP: inflammatory polyarthritis; RA: rheumatoid arthritis; FFQ: food frequency questionnaire; NHS: Nurses’ Health Study

**Table 2 nutrients-12-01456-t002:** Most important studies on fish oil supplementation on RA disease activity.

Study	Duration	Type	Dose gr/daily	Therapy	*N* Patients	Clinical Efficacy
Volker et al. (2000) [154]	15 weeks	Double-blind randomized trial	>2 gr	DMARDs	*	Improvement HAQ, MS
Adam et al. (2003) [155]	12 weeks	Double-blind crossover study	30 mg/kg	DMARDs	60	TJ, SJ
Berbert et al. (2005) [156]	12/24 weeks	Parallel randomized study	>3 gr, 9.6 gr olive oil	DMARDs	43	TJ, MS
Proudman et al. (2015) [157]	12 months	Double-blind placebo-controlled trial	5.5 gr	Triple therapy	140	Reduced failure of triple therapy
Kremer et al. (1995) [158]	48 weeks	Double-blind randomized trial	> 9 gr	NSAIDs	66	TJ, VAS physician
Miles et al. (2012) [159]	*	Systematic review (23 studies)	*	NSAID, DMARDs	*	TJ, SJ, MS, joint pain
Goldbeg et al. (2007) [160]	3–4 months	Meta-analysis (17 RCT)	*	NSAID, DMARDs	*	TJ, SJ, MS, joint pain, NSAID use
Park et al. (2013) [161]	16 weeks	Double-blind placebo controlled randomized trial	2.1 gr EPA + 1.2 gr DHA	NSAID, DMARDs	109	Decreased use of NSAIDs

TJ: tender joints; SJ: swollen joints; HAQ: health assessment questionnaire; MS: morning stiffness; NSAIDs: non-steroidal anti-inflammatory drug; DMARDs: disease modifying anti-rheumatic drugs; EPA: eicosapentaenoic acid; DHA: docosahexaenoic acid; * variable.

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
