# Peer review of "Dietary Habits and Nutrition in Rheumatoid Arthritis: Can Diet Influence Disease Development and Clinical Manifestations?"

_nutrients, 2020, doi:10.3390/nu12051456_

Round 1
Reviewer 1 Report
General impression: This is a review on a topic that is gaining importance in the clinical setting, since patients have become more concerned on the side effects of the drugs used for their treatment. It can be very useful for the rheumatologists to help them answer, at least in part, their patients’ questions on dietary recommendations in rheumatoid arthritis, based on existing evidence in the literature. However, I have some comments, please see below. I think a few fragments of the manuscript need rephrasing since they are difficult to follow.
Major points to review:
This is a very extensive topic and the authors have summarized literature findings related to RA risk followed by findings related to RA disease activity and progression. However, they mostly presented a list of findings, without drawing conclusions, when possible; for example, based on the presented data, what do the authors think about the role of vitamin D in RA development?
I think it would be very helpful for the reader if the authors included a figure with nutrients and their dietary sources that are involved in RA risk development (both protective and harmful) and the ones that are involved in disease activity and progression (beneficial and harmful). If preferred, 2 figures can be included, one for risk and one for the second part. This figure/s would represent a good summary of the review and maybe table 1 wouldn’t ne be necessary.
Also, the review includes very little information on nutrients that are considered to be harmful for inflammation. Besides red meat, patients also complain that they have flares or increased pain with the consumption of other foods, like tomatoes, eggplant, potatoes. Is there any evidence in the literature to support symptom worsening with these foods?
Related to red meat consumption, the authors should also mention the potential harmful effect due to its content in choline, a precursor of trimethylamine N-Oxide (TMAO), which has been related to inflammation. Also, TMAO is produced with the intervention of gut microbiome and Prevotella species produce it abundantly.
Alcohol – both the risk and disease progression paragraphs - do the presented studies check the correlation between RA and all types of alcohol or just wine? Please clarify.
The paragraph on the omega 3 and omega 6 fatty acids – EPA and DHA can both be synthesized by humans from alpha linolenic acid (ALA), by the intervention of the same enzymes that convert linoleic acid (LA) to dihomo-gamma linolenic acid and arachidonic acid (delta 5 and 6 desaturases). Actually, these enzymes have a higher affinity for ALA than LA as long as the intake of these fatty acids is at least 1:4 LA to ALA. But, dietary intake of ALA in the Western Diet is usually low compared to up to 30 fold higher intake of LA, which increases these enzymes’ preference to metabolize omega 6 polyunsaturated fatty acids (LA), thus the conversion of ALA is poor in humans and only a small proportion is converted to EPA and DHA. Moreover, the authors should emphasize that arachidonic acid (mention food sources also) is the precursor of pro-inflammatory eicosanoids, amongst which the classically described prostaglandins (PEG), especially PGE2, which has been related to the clinical signs of inflammation. EPA and DHA are the precursors of the specialized pro-resolving mediators, resovins, maresins and protectins, which are important in the resolution of inflammation.
Other comments
There are a lot of typos and sentences/phrases that difficult to follow.
Line 18 mechanisms
Line 52 represent both a disease risk and protective factor; replace “in base” of the properties– depending on the properties , or based on the properties
Line 57 primarily not primary
Line 63 is there really enough evidence to allow us to state that microbiome is strictly related to disease pathogenesis?
Line 73 the pro- or anti-inflammatory properties
Line 74 carbohydrates
Line 75 fats
Line 78 define abbreviations for TNF and IL
Line 81 define abbreviation for MMP3
Line 82 fruits, legumes, cereals, fish and olive oil
Line 83 diffuse?
Line 85 an omega3
Line 86 a high
Line 88 aliments -> foods
Line 90 took -> taken
Line 91 “could decrease CRP” – this is not an evidence of a direct anti-inflammatory effect. What are the exact results of the cited study?
Line 92 “could” again; does red wine reduce CRP in the cited study?
Line 93 also observed
Line 94 inhibit
Line 115 define ACPA abbreviation
Line 116 for instead of to more defined
Line 128 Omega
Line 133 due to the antioxidant
Line 141 context
Line 142 The authors only cite one study that found an association between coffee consumption and RA. The relation between RA and coffee is not clear, there are studies that found a protective effect of coffee. Please extend this part including more references.
Line 179 residing in the human
Line 181 again, I don’t think we can affirm that the gut microbiome is strictly related to inflammatory diseases
Line 195 this has only been shown to happen in animal models, it’s still a hypothesis, I think the authors should rephrase
Line 200 Reference 73 – the paper did not find an increased abundance of Prevotella or a decreased abundance of Bifidobacterium and Bacteroides
Line 197 – 215 This fragment of the manuscript is difficult to follow. The authors start by mentioning the quantitative changes in the microbiome that involve Prevotella and Bacteroides, then they mention Colinsella and then they return to Prevotella and Bacteroides again. They also repeat the fact the Prevotella is associated with RA. The part related to which foods changes in the abundance of the bacteria is also not clear and repeated. Please rewrite these fragment.
Line 127 Wester -> Western
Line 218 dietary patterns
Line 224 benefic -> beneficial
Line 225 Define TJ abbreviation
Line 226 benefic -> beneficial; effects -> effect
Line 227 though -> through
Line 228 despite -> Despite
Line 255 in preventing RA development
Line 263 1,25-dihydroxyvitamin D (1,25(OH)2D)
Line 269 activates the precursor 25-hydroxyvitamin D3 25(OH)D3 to 1,25(OH)2D; As a consequence
Line 272 vitamin D
Line 277 vitamin D, no capital letter required
Line 279 differences
Line 280 vitamin D, no capital letter required
Line 281 vitamin D, no capital letter required
Line 281, 282 based on the serum anti-citrullinated protein antibodies (ACPA) status
Line 284 of RA development
Line 286 showing on contrary an increases risk of RA -> showing the opposite, an increased risk of RA
Line 287 over than -> more than (or no than, just over)
Line 293 source -> sources
Line 295 limit -> limitation
Line 296 leads to use different cut-off in vary studies. -> leads to the use of different cut-off values across various studies
Line 298 for levels of 30-50 ng/mL [108]. Other cut-offs described are 20 ng/mL for deficiency and 20-30 ng/mL
Line 300-301 have been becoming matter of interest – are gaining interest?
Line 312 diseases; I think what the authors mean to say is that the observational studies found a protective effect of alcohol intake, however, these studies had several limitations, amongst which confounding factors and classification issues. This suggests the results of these studies must be interpreted with caution.
Line 313-314 There is no good quality randomized prospective clinical trial that has evaluated the relation between alcohol and the risk of RA development, so we can’t really state that “the evidence for a protective effect of alcohol on the development of RA may be stronger than in other diseases” and then say that “conclusive data are not available”. What could be said instead is that the observational studies suggest that alcohol could have a protective effect, although there are studies that had different findings.
Line 323 – cells’
Line 327-328 more studies are needed to investigate its therapeutic potential in humans [120].
Line 332 contest -> context
Line 333-335 “Together with a greater effective pharmacological treatment, patients are interested to all self-management strategies, able to improve symptoms for example resting, applying heat or cold attending physical therapy, and/or using a splint or brace.” This is difficult to follow, please rephrase
Line 335-336 A suggested dietary habit, what foods are recommended, and which should be avoided are some of the emergent questions that patients address to rheumatologists.
Line 337-338 “So, diet and its possible enhancing pharmacological treatment properties, became new matter of rheumatologist’s interest.” Please rephrase
Line 342 based on instead of on the base of
Line 348 – Humans can synthesize both DHA and EPA.
Line 353 also have
Line 359 effect
Line 373 foods
Line 385 On the contrary
Lin 392 IL6
Line 396 and decreased
Line 407 1 year
Line 410 also showed; I would recommend to substitute “showing” LDL….with a synonym
Line 412-418: First it’s mentioned that fasting decreases the number and activation of CD4+ cell and then it seems in RA fasting stimulates CD4+ activation and differentiation which is related with disease progression. Please clarify this.
Line 419: antioxidants
Line 412: with moderate to severe RA
Line 423-425 “Fasting followed by vegan diet or vegan diet could reduce symptoms and disease activity in RA changes, not only for microbiome modifications, but also for anti-inflammatory properties of ketone bodies during starvation.” This phrase is really difficult to follow, please rephrase.
Line 426 low vitamin D status, with low levels of serum 25(OH)D – what is after the coma is just repeating the same thing (low vitamin D status), unless the authors want to refer to D2 or D3.
Line 427 consider that disease severity
Line 433 define abbreviation for DAS
Line 440 “Data on alcohol and its effects on RA disease activity are controversial.” This sentence is just a repetition of the first sentence of the paragraph (line 438)
Line 442 On the contrary
Line 443 please define VAS and HAQ abbreviations
Line 446 must be
Line 447 potential “harm”?
Line 459-460 “Other studies did not observe improvement in serum lipids or in oxidative status in RA patients after daily supplementation of Lactobacillus casei 01”. What about changes in RA outcomes (tender joints, swollen joints and so on)?
Line 464 Please define SFA abbreviation
Line 465 “main”? meals
Line 466 yogurt
Line 467 The probiotic
Line 468-470 “No significant difference in effects on DAS28 and its components between ADIRA diet and Swedish diet but an improvement in disease activity during intervention period were registered.” DAS28 is a measure of disease activity in RA. So, were there any effects of the dietary intervention on disease activity or not? Please clarify.
Line 476 foods
Line 480 drinks
Line 484 context
Line 493 The evidence of diet’s impact on RA disease activity
Line 495 the importance of defining
Line 496 Please define abbreviation DMARD; could be enhanced by adding
Line 502 help in reducing RA disease activity, delaying disease progression and probably decreasing dose of drugs
Figure 1: Diet role in pathogenesis – even in the abstract, the authors state that diet probably has a protective effect on the risk of RA development, however the figure does not reflect that.
Author Response
Dear Reviewer,
many thanks for your check and suggestions. Here a summary of the changes in the manuscript:
- We conclude the paper with the section "Dietary recommendation for RA patients". Following your suggestion, we specified the role of vitamin D also in RA prevention.
- We produced as you suggested a new figure, summarizing all the nutrients and food sources that have an effect on the disease development and progression. Considering that dietary evidence are substantially overlapping between food helpful/harmful in both disease development and progression, we decided to produce only one figure. Moreover we think the figure helps the reader to get practical conclusions from the paper. Accordingly, we eliminated table 2
- We extended the sections about harmful food, such as milk, tomatoes and potatoes
- We included in the main text, the topic of meat derived choline and TMAO, including also their production by Prevotella
- We clarified the role of Alcohol, stating specifically also about red wine
- We extended the section about eicosanoids metabolism, including the unbalance between omega3 and omega 6, and the role of EPA and DHA in the resolution of inflammation
- We rewrote the section about the correlation of dysbiosis with the pathogenesis of the disease, making it more readable
- We extended the section about the coffee consumption, including the possible bias related to smoking
- We extensively rewrote the sentences you pointed as difficult to follow
- We included your suggestion about the second Alcohol and vitamin d sections as well as probiotics section
- Many thanks for the exteded typos and spell check. We providet to correct them
- As you suggested we included the potential harmful effect of the diet in the abstract and redrew the figure 1
We hope this revised version would suitable to you.
Best Regards
The Authors
Reviewer 2 Report
Congratulation for your paper. There is a few comments related to your manuscript.
Line 42 to line 50 does not correspond to “Diet as a risk factor”, that would be more “Genetic factors” but is a different topic. I suggest mention it briefly in the introduction.
Line 108 “RA patients had a higher intake of fish, potatoes, mushrooms and organ meats”, This is good or bad regarding to inflammation? I think for this line you can express better what are the implication of the higher intake of fish etc. and another point to discuss here is the type of population, and the same in the Benito-Garcia et al on line 109. For all this “dietary habits” subtitle could be very helpful to show a table comparing the different population with their dietary habit and the clinical outcomes of RA, this is with the purpose to make it more visually comparable.
I strongly suggest to reference this paper: “Design of an anti-inflammatory diet (ITIS diet) for patients with rheumatoid arthritis.” Contemp Clin Trials Commun. 2020 Jan 21;17:100524. doi: 10.1016/j.conctc.2020.100524. This paper shows the beggining to understand how diet modify the behavior of the RA, which is the objective of your review.
There is another review which touch important topics related to your review. This is “Circulating Pro- and Anti-Inflammatory Metabolites and Its Potential Role in Rheumatoid Arthritis Pathogenesis.” Cells. 2020 Mar 30;9(4). pii: E827. doi: 10.3390/cells9040827. This paper has a part where they discuss the diet and the metabolites content on the food, which could be pro or anti-inflammatory.
Author Response
Dear Reviewer,
many thanks for your check and suggestions. Here a summary of the changes in the manuscript:
- As you suggested, we moved the "genetic factor" in the introduction
- We produced a new table summariziong the characteristic of populations enrolled, the dietary habits and the outcomes
- We checked both the papers you pointed, and included a reference to them into the main text
We hope this revised version would suitable to you.
Best Regards
The Authors
Round 2
Reviewer 1 Report
The authors have done a great job with all the suggestions. Just a few minor things:
Line 58: concerning- not appropriate word
Line 123: please clarify what sweet meet is
Line 132 – due to its flavonoid content; which inflammatory marker can be decreased by black tea?; leukocytes
Line 293 content
Line 302: 0.21g/day
Line 622: delete it (from which it they can migrate)
Line 634: an increase of related Th17 cytokines
Line 635: increased gut permeability
Line 687: the word "analysis" is used twice, please replace one of them
Line 803: chronic diseases
Line 804: the authors meant that a definitive indication can’t be “made”?
Line 805: must be taken (delete to)
Line 895: which the large content in polyphenols can reduce inflammation, - with a large content in polyphenols that can reduce
Line 823: interested in
Line 828: I think the word :rising is not necessary
Line 829: dietary intake
Line 830: dietary intake
Line 870: derived from
Line 870: have, on the contrary, mainly
Line 947: by inhibiting
Line 966: reported that
Author Response
Dear Reviewer,
thank you again for the revision. We provided the requested spell/form changes.
All the best
The Authors